# Investigating the Self-Reported Health Status of Domestic and Overseas Chinese Populations during the COVID-19 Pandemic

**DOI:** 10.3390/ijerph18063043

**Published:** 2021-03-16

**Authors:** Zhuxin Mao, Bohao Chen, Wei Wang, Paul Kind, Pei Wang

**Affiliations:** 1School of Insurance, Southwestern University of Finance and Economics, Chengdu 611130, China; zhuxinmao@swufe.edu.cn; 2School of Public Health, Fudan University, Shanghai 200032, China; 20211020054@fudan.edu.cn; 3Institute of Health Sciences, University of Leeds, Leeds LS2 9NL, UK; p.kind@leeds.ac.uk

**Keywords:** COVID-19 pandemic, self-reported health status, China, population health, EQ-5D

## Abstract

To control the spread of COVID-19, governments in different countries and regions implemented various types of lockdown and outdoor restrictions. The research aimed to describe and compare the health status of Chinese people both domestically and abroad in this global health crisis. An online questionnaire survey was distributed to Chinese mainland citizens living in Hubei (the lockdown province), outside Hubei, and those living abroad in 2020. A total of 1000 respondents were recruited and reported worse health status compared with Chinese population norms. People living in Hubei reported worse health status than those living outside Hubei but revealed better health status than overseas respondents. It was clear that the pandemic as well as strict lockdown and outdoor restriction policies affected Chinese people’s health. It is important for the Chinese government to be aware of the negative impact of such strict policies and to take measures to reduce the panic of society when implementing similar policies in the future. It also implies that governments in other countries should promote social support for those who live far from home and actively call for support for non-discriminatory attitudes toward ethnic minorities.

## 1. Introduction

Since COVID-19 first emerged in Wuhan, China, at the end of 2019, the virus has spread to more than 200 countries and territories around the world. The outbreak has been declared as a pandemic by the World Health Organization, and millions of COVID-19 cases have been reported, causing tens of thousands of deaths. Given the novelty of this virus, many epidemiological and clinical features of COVID-19, including “the extent of infection, the route of transmission, the full range of disease presentation, and the viral dynamics”, remain unknown [1], despite substantial efforts that have been made by scientists all over the world. What we do know is that the COVID-19 pandemic is an unprecedented global crisis in this modern age and has largely changed people’s lives and livelihoods.

In response to the COVID-19 outbreak, on 23 January 2020, the Chinese government implemented lockdown policies in Wuhan [2], which is the capital city of Central China’s Hubei province and where the virus was first identified. Soon after the lockdown in Wuhan, similar policies were imposed on other cities in Hubei province to tackle the spread of the virus. The epidemic’s initial spread throughout China coincided with the Chinese New Year period with large-scale personnel movement, which caused great challenges for the prevention and control of the epidemic. The Chinese government at all levels took control measures to close all access routes from and to Hubei [3]. As the economic center of central China, Wuhan in Hubei plays a pivotal role in the country’s daily economic and trade logistics. The Chinese government’s decision of closing the transportation in Wuhan, Hubei, at that time reflected the firm belief of the Chinese government to effectively contain the spread of the epidemic and protect the lives and health of residents. As for the other provinces in China, outdoor restrictions were implemented to handle the crisis. The government called on citizens to go out as little as possible and to wear masks when they went out. Except for supermarkets and pharmacies, large public places, including shopping malls and catering and entertainment venues were closed nationwide. Mass gatherings including sports events and examinations were canceled or postponed. Under such lockdown policies and outdoor restrictions, the amount of time people stayed outdoors was cut down, and the frequency of people leaving their homes was limited. It means that people had fewer social contacts with other people outside their household than normal. Those policies were believed to be crucial for stopping the spread of the virus [4], but at the same time, they could lead to social isolation, which may cause loneliness, anxiety, and depression, affecting the life of individual people as well as the whole society in many aspects [5,6]. With a higher risk of infection from the disease itself and the most restrictive measures imposed by the lockdown policy, people living in Hubei were likely to suffer a higher level of pressure than those outside Hubei, which was likely to affect their health status and generate specific needs for healthcare.

As the virus continues to spread to other countries and territories around the world, overseas Chinese populations can be under a situation of double pressure from both Chinese society and foreign societies [7]. On the one hand, they face similar difficulties caused by the increasingly serious epidemic as those domestic Chinese citizens, including similar lockdown or restriction measures taken by their local governments, a shortage of daily necessities, and a lack of social interaction. In addition, many of the overseas Chinese citizens, for example, Chinese students studying abroad, may also feel excluded from the Chinese society, when they were discouraged from returning to their home country by the verbal abuses on Chinese social media as well as the international travel policies issued by the Chinese government [8]. On the other hand, they received stigmatized expression, discrimination, and hatred toward Chinese ethnics in foreign societies [9]. Clearly, living aboard caused specific challenges and issues that could add to the negative impact of the pandemic on Chinese nationals.

Among the many factors that may be associated with the health status of people during the COVID-19 pandemic, the contribution of regions has yet to be explored. In the early stage of the pandemic, staying in Hubei, outside Hubei, and overseas represented three distinctive scenarios with different levels of living pressure, that may affect the perceived health status of Chinese residents differently and thus generate different healthcare needs. The type of healthcare and social support required by those living in/outside Hubei or aboard may differ substantially from one another. Thus, there is a need to investigate whether a regional difference exists in health status among Chinese populations. This information is critical in designing region-specific support measures as part of the national contingency plan for public health emergency response. Studies have been conducted to evaluate the impact of COVID-19 on people’s psychological well-being since the outbreak of the novel coronavirus epidemic [10,11], while limited studies have examined individuals’ self-reported health status as a whole. One newly published paper has surveyed the overall health status of the Chinese general population in the pandemic [12], however, since that study recruited participants from only one Chinese city, which was not widely affected by COVID-19 (only eight definite COVID-19 cases were identified in the city selected by the authors), it may not represent the health status of Chinese people in other seriously affected regions well. To the best of our knowledge, there has been no existing literature comparing the health status of Chinese populations in different regions in the context of the pandemic, even though people in different regions were under different lockdown or restriction policies and were under various levels of pressure. Hence, we conducted this study to describe, analyze, and compare the health status of Chinese people both domestically and abroad in this global health crisis.

## 2. Materials and Methods

### 2.1. Study Design and Participants

In this cross-sectional study, we distributed an online questionnaire survey via the WeChat platform to Chinese mainland citizens living in Hubei (the lockdown province) and outside Hubei between February and March 2020. The vast majority of new cases in China occurred between February and March 2020, which can be considered as the worst period of the pandemic in China [13,14]. We distributed the questionnaire to Chinese citizens living abroad between mid-March and April 2020. The overseas epidemic situation began to deteriorate sharply in mid-March. Taking America and Europe as examples, the cumulative number of new cases on 15 March was 2655 and 50,730, respectively, and on 30 April, they surged to 1,246,190 and 1,448,952, respectively [15]. We selected this period to collect data of overseas Chinese populations to make the data more comparable to that of Chinese mainland respondents.

We used a snowballing strategy in our data collection, because it was the most cost-effective way for us to obtain a large sample size to address our research aim. We distributed an electronic questionnaire to individuals who met the requirements of three different geographical distributions through WeChat and encouraged them to distribute the questionnaire to their contacts who also met the inclusion criteria. In the data collection, we tried our best to deliver our questionnaire to individuals who could form a sample to reflect the characteristics of the target population. Because of the nature of the COVID-19 pandemic, we were not able to collect data using a paper questionnaire to avoid face to face contact with respondents. The online questionnaire survey was mainly distributed through WeChat, which is one of the most popular multipurpose messaging and social media apps in China.

### 2.2. Survey Questions

Our survey mainly consisted of two parts.

EQ-5D-5L (the 5 level version of the EQ-5D questionnaire) formed the first part of our survey and was used to collect data on self-reported health status. The Chinese versions of EQ-5D have been widely applied to various health studies including general population studies and patient-specific studies in China [16,17]. A diverse range of validation studies for the Chinese version of EQ-5D has been conducted [18,19]. EQ-5D was valid and reliable to be applied in China, based on satisfactory statistical results produced.

EQ-5D (5L) contains a descriptive system and a visual analogue scale (EQ-VAS) to record a respondent’s self-rated health status on the day of survey (“today”) in order to minimize recall bias. The descriptive system can describe individuals’ health status by constructing a five-dimensional (mobility, self-care, usual activities, pain/discomfort, anxiety/depression) health profile, with each dimension being divided into five levels (no problems, slight problems, moderate problems, severe problems, extreme problems). The five-dimension and five-level descriptive system generates a total of 3125 health states, each of which can be referred to as a five-digit code. For example, state “11111” indicates no problem on all five dimensions. Each health state can also be converted into a health utility score through a corresponding value set. The 5L value set based on Chinese residents is available in published literature [20], which provides information about how the value set was derived and the scoring algorithm for computing utility scores. The EQ-VAS records the individual’s overall self-rated health on a vertical visual analogue scale, taking values between 0 (worst imaginable health) and 100 (best imaginable health).

The second part of our survey recorded respondents’ sociodemographic information including age, gender, education, marital status, and information about existing chronic disease.

### 2.3. Data Analysis

EQ-5D population norm data have often been classified by age and gender for reporting in previous population studies [21,22]. In our study, descriptive analyses by region were also stratified by gender and age groups. Age groups were categorized into three groups: under 40 years, 40–60 years, above 60 years. We used chi-squared tests where appropriate to test cross-region differences in demographic variables. Regional differences in self-reported health status were analyzed by showing percentages of people reporting any problems in each EQ-5D dimension. After computing utility scores according to the scoring algorithm [20], mean utility scores and mean VAS scores were calculated by region (Hubei, outside Hubei, overseas respondents) and were compared across regions. We used outside Hubei as the reference category for comparisons. Dummy variables were generated for each regional categorization. Multiple regression models were constructed to estimate how health status (percentages of respondents reporting problems in each dimension of EQ-5D, utility scores, and VAS scores) varied in different regions, controlling for demographic indicators. The first set of models only controlled for gender and age; then education, marital status, and chronic disease status were additionally controlled. The significance level used in our statistical tests and regression analyses was set at 5%.

All analyses were conducted in SPSS version 23 (IBM Corp, New York, NY, USA).

## 3. Results

A total of 1000 respondents were recruited in this study during the survey period. The demographic characteristics of the respondents are shown in Table 1. Among them, 229 were in Hubei province, 509 were in other provinces outside Hubei, and 262 were from overseas countries/regions. Respondents from the three regions differed significantly in age, education, marital status, and chronic disease status. The overseas group had a higher proportion of young (under 40 years), higher educated (postgraduate degree and above), unmarried, and non-chronically ill respondents compared to the other two groups, while the Hubei group had a higher proportion of elder (above 60 years), less educated (college and below), married, and chronically ill respondents. In general, female, younger, highly educated people were more likely to participate in this survey.

In total, 46.5% of the respondents reported their health as “11111”. The percentages of respondents reporting “no problems” were: 90.2% for mobility, 98.9% for self-care, 88.9% for usual activity, 84.7% for pain/discomfort, and 55.2% for anxiety/depression. As a result, the mean utility score was 0.93 (standard deviation (SD): 0.12). The mean EQ-VAS was 87.8 (SD: 41.2). Percentages of respondents reporting any problems in each EQ-5D dimension, which were stratified by region, gender, and age group are presented in Table 2. Respondents living in Hubei province or overseas reported more problems in all EQ-5D dimensions than respondents living outside Hubei in almost all age groups. More specifically, male respondents younger than 40 years who lived in Hubei reported more problems in mobility and self-care than those in other regions; while male respondents younger than 40 years who lived overseas reported more problems in usual activities, pain/discomfort, and anxiety/depression. As for female respondents, those younger than 40 years who lived overseas reported more problems than those in Hubei province, who in turn reported more problems than those in China but outside Hubei. For respondents older than 40 years, respondents from Hubei reported more problems in all EQ-5D dimensions than respondents from other regions.

The mean utility and VAS scores were calculated by region and are presented in Table 3. For respondents younger than 40 years old, those who lived in mainland China had higher mean utility and VAS scores than those who lived overseas. While for respondents older than 40 years, those who lived in Hubei generally had the lowest mean scores compared to respondents who lived outside Hubei and overseas.

The likelihood of having any problems in each EQ-5D dimension and the variation of utility and VAS scores were analyzed using regression models. The first set of models was controlled for age and gender and is presented in Table 4. It shows that respondents in China but outside Hubei province had better health status (with lower odds of having any problem in EQ-5D dimensions) when compared with respondents in Hubei or overseas. Respondents living in Hubei had a significantly higher odds ratio for mobility. Odds ratios for respondents living in Hubei and overseas were also significant in usual activities and anxiety/depression. As for utility and VAS scores, respondents living in China but outside Hubei had significantly higher scores than those in Hubei and overseas.

Having further controlled additional variables including marital status, level of educational attainment, and chronic disease condition, respondents in China but outside Hubei province still had better health in terms of EQ-5D states, utility, and VAS scores than the rest of the respondents (Table 5). They had lower odds of having any problem in almost all EQ-5D dimensions and higher utility and VAS scores. Although the results were similar to that in Table 4, some of these indicators became insignificant.

## 4. Discussion

Our study used the EQ-5D to investigate the self-reported health status of Chinese populations domestically and abroad during the worst period of the COVID-19 pandemic. Despite recruiting a relatively younger and highly educated dominant sample compared to the structure of the Chinese general population [23], the “no problems” rates of EQ-5D for mobility, usual activities, and anxiety/depression as well as the mean utility score in our study were lower than the 5L Chinese population norms published previously [24]: 90.2% vs. 94.4% for mobility, 88.9% vs. 95.45% for usual activities, 55.2% vs. 73.2% for anxiety/depression, 0.93 (SD: 0.12) vs. 0.96 (SD: 0.07) for mean utility scores. It has been consistently agreed in the literature that young, more educated people tended to report better EQ-5D status [22,24,25]. With a large proportion of young and highly educated respondents, our study should have reported better health status. This implies that COVID-19 as well as the relevant lockdown or restriction policies affected people’s self-reported health.

Compared to the newly published paper that also surveyed a Chinese general population in Shanxi province using the 3 level version of EQ-5D, our study also reported worse health status in terms of the “no problems” rates of mobility (90.2% vs. 96.1%), usual activities (88.9% vs. 98.1%), and anxiety/depression (55.2% vs. 82.4%) as well as mean utility scores (0.93 vs. 0.95) [12]. On the one hand, our study used the 5L, which was believed to be more sensitive in detecting health problems and have less ceiling effect compared to the 3L [24,26]. Moreover, we collected data from Hubei residents who were affected by strict lockdown policies as well as overseas Chinese people who received additional pressure from foreign societies, which can in turn caused negative influences on their health. As these respondents from Hubei and overseas tended to report worse health status than those from other Chinese provinces, as presented in our study, this may explain why our study sample reported worse health status compared to the single-city study conducted by Ping and colleagues [12].

Among respondents living in China, people living in Hubei province reported worse health status than those living outside Hubei. This indicates that the COVID-19 pandemic as well as the lockdown policies relevant to Hubei were seriously affecting people in Hubei, wherein their self-reported health was adversely affected to a larger extent. People who were living in Hubei may have had more challenges in terms of a shortage of food supplies and the lack of social interactions during the lockdown period, which may have made them develop post-traumatic stress disorder. They may also have experienced sorrow on the passing away of friends, relatives, or someone close, which can have prolonged negative effects. As for overseas respondents, their health status was reported to be even worse than that of Hubei respondents in some EQ-5D dimensions. A large number of them (59.5%) reported anxiety/depression, while the average health utility score of respondents from overseas areas was 0.91, which is close to the utility score of Chinese residents over the age of 71 in the Chinese population norm data [24]. This reflects the substantial impact of the pandemic on the health of overseas people. It may be because they not only had to face difficulties caused by the pandemic including a shortage of daily necessities as well as the lack of social interaction, but they may also have received or been afraid of receiving stigmatized expression, discrimination, and hatred toward Chinese ethnics in foreign societies. All these could increase overseas Chinese people’s anxiety/depression, which in turn may have led to a decrease in self-assessed health in general. Another reason may be that most of the overseas respondents were in the young age group, which was likely to have access to social media. They tended to witness the strict lockdown or outdoor restrictions implemented by the Chinese government, which had effectively controlled the spreading virus within China at the time when they were completing the survey. While they also found that their local governments were less strict in terms of announcing “hard” lockdown and outdoor restriction policies while the pandemic was rapidly spreading. For example, in the UK, the prime minister first announced the “herd immunity” strategy without taking effective actions to control the virus spreading in early March 2020. The strategy was only changed later to ask citizens to stop non-essential contact with others and to stop all unnecessary travel. This may also have increased their psychological burden and deteriorated their health.

Contradictory to previous studies [21,24], our results showed that people with a younger age tended to report more health problems (especially in usual activities and anxiety/depression) and thus receive a lower utility score. It may be because working and studying tended to be an important component of the daily life among the younger respondents, but since they had to stay home due to the pandemic, their daily activities were greatly affected; while for the elderly, there may be less demand for going out, thus their usual activities were less affected. In addition, the younger age group was more likely to be exposed to social media and had a higher level of awareness of the pandemic than the older age groups, which can bring the younger group more mental pressure and burden.

Despite the fact that the spread of the virus in the mainland China had been quickly and effectively controlled, it seems that the extremely strict lockdown policies in Hubei had negatively affected the health status of the local residents. Residents in regions with a more restrictive policy were found to be associated with worse health status, than those in areas with a less restrictive policy. Therefore, it is important for the government to consider providing more social support, especially in the mental health aspect, to local residents, while imposing a more restrictive containment policy to a specific high-risk region in response to future public health crises. On the other hand, our overseas Chinese respondents who reported the worst health status may represent another vulnerable group that needs more help and understanding. Continuous efforts are required from every nation and community to eliminate the stigma against residents from a particular country or area that is being affected by a public health event. It is also suggested that the embassy may collaborate with local governments to provide citizens abroad with timely access to more psychological counseling services.

Our study has some limitations. First, we collected our data using an online survey with a non-probability sampling strategy, and thus certain selection bias was inevitable. As a result, we obtained an unbalanced sample dominated by highly educated young people and cannot represent the Chinese general population. Second, at the time when we designed the questionnaire survey, we chose to collect age information by three age group (under 40 years, 41–60 years, and above 60 years), without obtaining their exact ages, which restricted our statistical analysis. With an exact age of each individual, our linear regression analysis could have been more efficient. Third, during the public health emergency of the COVID-19, people’s self-assessed health status may have changed substantially along with any current information they received about the virus, therefore the data we collected in this study can only reflect the health status of the respondents at a specific time point, which may have been fluctuating from time to time. While a longitudinal study is more likely to suggest cause-and-effect relationships by virtue of its scope, a cross-sectional design was deemed more practical in the face of a pandemic and suited our study objective, which was to obtain a timely and descriptive snapshot of health status among people living in different regions during such a special time period. Fourth, given that different countries responded to the pandemic very differently, especially in the early stage of the outbreak, some countries might impose tighter control measures than others. We are aware that the perceived health status may vary among countries where overseas Chinese populations were staying. However, due to the small sample size, we were unable to perform any meaningful subgroup analysis based on country of residence. While our overseas respondents may form a good representation of young Chinese nationals who are largely colleague students and professionals seeking better prospects in developed countries, their perceptions cannot be readily generalizable to those living in other areas such as Africa.

## 5. Conclusions

In this study, the EQ-5D was found to be useful to describe health status in the COVID-19 pandemic. This study provides evidence on the variations of self-reported health status among people living in different regions during the COVID-19 pandemic. The study showed that the pandemic as well as strict lockdown and outdoor restriction policies affected Chinese people’s health, especially mental health, to some extent. People living in Hubei reported worse health status than those outside Hubei, while Chinese nationals residing abroad appeared to have the worst health status. Region-specific social support measures especially in mental health should be considered as part of the national contingency plan for public health emergency response.

## Figures and Tables

**Table 1 ijerph-18-03043-t001:** Characteristics of respondents in the three regions.

Characteristic	Outside Hubei, n (%)	Hubei, n (%)	Overseas, n (%)	*p* Value
**Gender**				0.19
Male	180 (35.7)	97 (42.4)	101 (38.6)	
Female	329 (64.6)	132 (57.6)	161 (61.5)	
**Age** (years)				<0.01
≤40	330 (64.8)	120 (52.4)	230 (87.8)	
41~60	157 (30.8)	64 (28.0)	25 (9.5)	
>60	22 (4.3)	45 (19.7)	7 (2.7)	
**Education level**				<0.01
Postgraduate degree and above	161 (31.6)	60 (26.2)	192 (73.3)	
Undergraduate	191 (37.5)	85 (37.1)	48 (18.3)	
College and below	157 (30.8)	84 (36.7)	22 (8.4)	
**Marital status**				<0.01
Unmarried	264 (51.8)	67 (29.3)	170 (64.9)	
Married	223 (43.8)	149 (65.1)	84 (32.1)	
Divorced/widowed	22 (4.3)	13 (5.7)	8 (3.1)	
**Chronic diseases**				0.01
No	444 (87.2)	194 (84.7)	242 (93.1)	
Yes	65 (12.8)	35 (15.3)	20 (6.9)	

**Table 2 ijerph-18-03043-t002:** Percentage of respondents reporting any problems in each EQ-5D dimension, by age group, gender, and region.

EQ-5D Dimension	≤40 Years			41~60 Years			>60 Years		
	Outside Hubei	Hubei	Overseas	Outside Hubei	Hubei	Overseas	Outside Hubei	Hubei	Overseas
Male	119	53	92	50	26	7	11	18	2
Female	211	67	138	107	38	18	11	27	5
Mobility									
Male	10.9%	17.0%	16.3%	6.0%	19.2%	0	0	5.6%	0
Female	8.1%	7.5%	9.4%	5.6%	13.2%	5.6%	0	18.5%	0
Self-care									
Male	1.7%	3.8%	2.2%	0	0	0	0	0	0
Female	0.9%	0	0.7%	0.9%	2.6%	0	0	0	0
Usual Activities									
Male	10.1%	13.2%	25.0%	6.0%	19.2%	0	0	0	0
Female	7.6%	11.9%	17.4%	4.7%	7.9%	22.2%	0	3.7%	0
Pain/Discomfort									
Male	10.9%	11.3%	16.3%	10.0%	15.4%	0	27.3%	22.2%	0
Female	14.2%	9.0%	15.2%	16.8%	21.1%	22.2%	36.4%	37.0%	40.0%
Anxiety/Discomfort									
Male	43.7%	47.2%	58.7%	20.0%	50.0%	14.3%	27.3%	44.4%	50.0%
Female	46.4%	41.8%	66.7%	23.4%	47.4%	44.4%	27.3%	33.3%	0

**Table 3 ijerph-18-03043-t003:** Mean EQ-5D-5L utility and visual analogue scale (VAS) scores, by age group, gender, and region.

Utility/VAS	≤40 Years			41~60 Years			>60 Years		
	Outside Hubei	Hubei	Overseas	Outside Hubei	Hubei	Overseas	Outside Hubei	Hubei	Overseas
Male	119	53	92	50	26	7	11	18	2
Female	211	67	138	107	38	18	11	27	5
Utility *									
Male	0.93	0.91	0.9	0.98	0.9	0.99	0.97	0.95	0.98
Female	0.94	0.93	0.91	0.97	0.92	0.95	0.96	0.92	0.98
VAS **									
Male	90.4	89.6	85.4	90.4	84.0	91.4	90.5	92.1	100
Female	87.9	89.4	84.2	89.9	90.1	85.7	88.6	76.3	88.6

* mean utility score = 0.93 with a standard deviation of 0.12; ** mean VAS = 87.78 with a standard deviation of 14.32.

**Table 4 ijerph-18-03043-t004:** Multiple logistic regression analyses on having any problems in EQ-5D dimensions and multiple linear regression analyses on EQ-5D-5L utility and VAS (visual analogue scale) scores, by region, controlled for sex and age group.

Region	Mobility	Self-Care	Usual Activities	Pain/Discomfort	Anxiety/Depression
	OR (95% CI)	OR (95% CI)	OR (95% CI)	OR (95% CI)	OR (95% CI)
Outside Hubei *	1	1	1	1	1
Hubei	**1.89 (1.12–3.17)**	1.53 (0.36–6.50)	**1.80 (1.04–3.12)**	1.00 (0.62–1.54)	**1.47 (1.06–2.05)**
Overseas	1.40 (0.83–2.34)	1.01 (0.23–4.38)	**2.92 (1.83–4.67)**	1.24 (0.81–1.90)	**2.11 (1.54–2.89)**
	Utility	VAS			
	Estimate beta	Estimate beta			
Outside Hubei *					
Hubei	**−0.03 (−0.05~−0.01)**	−1.36 (−3.63~0.91)			
Overseas	**−0.03 (−0.05~−0.01)**	**−4.03 (−6.19~−1.88)**			

OR: odds ratio; Bold ORs indicate significant differences. * Reference category: outside Hubei.

**Table 5 ijerph-18-03043-t005:** Multiple logistic regression analyses on having any problems in EQ-5D dimensions and multiple linear regression analyses on EQ-5D-5L utility and VAS (visual analogue scale) scores, by region, controlled for sex, age group, education level, marital status, and chronic diseases.

Region	Mobility	Self-Care	Usual Activities	Pain/Discomfort	Anxiety/Depression
	OR (95% CI)	OR (95% CI)	OR (95% CI)	OR (95% CI)	OR (95% CI)
Outside Hubei *	1	1	1	1	1
Hubei	**2.10 (1.24–3.56)**	1.83 (0.42–8.00)	**2.05 (1.17–3.60)**	0.99 (0.62–1.57)	**1.57 (1.12–2.20)**
Overseas	1.01 (0.46–2.27)	0.29 (0.04–2.26)	1.83 (0.84–4.03)	0.91 (0.49–1.71)	1.23 (0.76–2.01)
	Utility	VAS			
	Estimate beta	Estimate beta			
Outside Hubei *					
Hubei	**−0.03 (−0.05~−0.02)**	−1.51 (−3.80~0.78)			
Overseas	**−0.03 (−0.06~−0.01)**	−0.93 (−4.23~2.37)			

OR: odds ratio; Bold ORs indicate significant differences. * Reference category: outside Hubei.

## Data Availability

Data are available from the authors upon reasonable request.

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
