# Peer review of "Investigating the Self-Reported Health Status of Domestic and Overseas Chinese Populations during the COVID-19 Pandemic"

_ijerph, 2021, doi:10.3390/ijerph18063043_

Round 1

Reviewer 1 Report

The study is very interesting and contributes to knowledge, I would recommend addressing the observations that I dare to propose to give greater clarity to the manuscript

Reviewer 2 Report

Dear Authors, you have presented self-reported health status in Chinese population during COVID-pandemic, but I have some comments about it.

Could you explain how did recruit the participants?

In section Statistical Analysis add more information about what kind of test did you use and what for, what about p-value?  

Table 1 is this necessary to present variable such as male, female, divorced as bold.

  1. 4 – you present data in text but where I can find them in which tables.

Discussion could you extended, for example of results of other studies before SARS-CoV-2 pandemic.   

Conclusion is more like summary which should be presented in discussion, because you give possible answers to you results.

Present conclusions based on your research.

Reviewer 3 Report

Many thanks for giving me the opportunity to review the manuscript entitled "Investigating the self-reported health status of Chinese populations at domestic and abroad during the COVID-19 pandemic".

This study was focussed to investigate the self-reported health status of Chinese populations at domestic (Hubei and outside Hubey in China) and abroad during the worst period of the COVID-19 pandemic. The data on self-reported health were collected via the internet using a five-item scale. A total of 1000 responses of Chinese people were received, and such sample size is adequate for statistical analysis. Statistical analysis is based on descriptive analysis by showing percentages of people reporting any problems in each scale dimension, logistic regression to estimate risk of health problems, and mean utility scores and mean scores of visual analogue scale (VAS). The authors argue that the pandemic as well as strict lockdown and outdoor restriction policies negatively affected Chinese people’s health. People living in Hubei reported worse health status than those living outside Hubei but reported better health status than overseas respondents. The findings from this study may be useful to readers who interested in COVID-19 pandemic consequences. Especially they are important for the government leaders to be aware of the negative impact of strict policies and to take measures to reduce the panic of society when implementing similar policies in the future.

The article is written in a typical format. The overall rationale, design and objectives of the data analysis conducted in this study are presented clearly and concisely. There is an excellent description of the study design, sampling frame and data collection methodology for the dataset for this study.

Nevertheless, there are some drawbacks that I recommend to fix:

  1. The authors argue that the main cause of health deterioration is solely strict quarantine restrictions. In fact, the reasons for this are broader. The authors should discuss this in more detail.
  2. Data analysis, line 114: "mean utility scores and mean VAS scores were calculated". I recommend a more detailed explanation of these methods with examples. A reference to a literature source where these methods can be found in more detail is also recommended. Consequently, the meaning of corresponding estimations that are presented in Tables 4 and 5 should be explained.
  3. References must be presented according to the requirements of the journal.

Thank you for considering my opinion.

Reviewer 4 Report

Comment to the Author

Manuscript ID: ijerph-1117128

Title: Investigating the self-reported health status of Chinese populations at domestic and abroad during the COVID-19 pandemic

Page1, line16

The word "reported" is used in the same sentence. You need to change your explanation.

Page1, line43

Can you please provide a citation for this as well as the previous one? It only lists the national policies but does not describe the core background.

Page2, line52

Who thinks it is important?  Is it the author? Is it a reference? You should look deeply into the previous studies.

Page2, line55

I think there are better references than [2]. Please examine them carefully.

Page2, line56

Where do you mean by "other countries"? Can you really say that China was the only country that had a policy?  I think other countries had policies, too. There is no comparison of China's policies with those of other countries in the text.

So, I am not sure.

Page2, line58

Is this a personal fantasy of the author?

Page2, line58

This part is the original part and the core of this research, but the explanation is only speculation and is insufficient. Please state it logically.

Page2, line65-68

I feel that this sentence is inconsistent with the sentence (line 58) that was under pressure from Chinese society. Please correct the expression to clarify the purpose of your research.

Page2, line69

When did you say "recent studies"? Since the pandemic, right? I don't feel comfortable with the sentence. You should be "Fool-proof English".

Page2, line 76

Which is correct, little or no?

If it is little, what are the reports of global infections like COVID-19?

Page2, line78

From the flow of the introduction, I don't understand how you arrive at this goal.

Page2, line79

As the situation regarding coronas is constantly changing, it is possible that the situation described in the introduction may have changed in the present. It would be good if you could add the significance of comparing the health status of Chinese citizens in the early stages of the spread of infection and present the results.

Page2, line79

The content currently described in the introduction makes it unclear the need to compare the health status of domestic and overseas residents. Please describe clearly.

Page 2, line 83

The number of respondents to the WeChat survey was 1,000 as a result. But how did people find out about the survey in the first place? I have the impression that the characteristics of this target are unbalanced, but did you have any problems with your advertising methods?

Page2, line 83

How many Chinese people living in Hubei, in mainland China outside Hubei, and living abroad were affected by COVID-19?

Page2, line 83

In which country or region do “Chinese citizens living abroad” refer to people living in? If the country or region is different, it will affect the results.

Page2, line 83

How is the penetration rate of WeChat in China? It is easy to imagine that it is popular among the young, but is it also widely used among the elderly? Doesn't this lead to bias in the target audience?

Page2, line83-87

It would be better explain why you chose February and March as the survey period for Hubei and March and April as the survey period for Chinese living abroad, using specific numbers of infected people and other evidence. Subjective description (worst period of the pandemic) is not enough. Even now, the situation is changing every moment.

Page2, line92

Are these Questionnaire interpreted as listening to what happened after the pandemic? Or were the respondents in these health conditions before the pandemic? Moreover, what kind of pain is it? (Musculoskeletal disease or mental). There is not enough detail.

Page 3, line102

I think that there is a lack of explanation of how the health utility score is calculated.

Page3 128-132

In situations where there are differences between these groups, it is inappropriate to compare outcomes. At least in terms of age and gender, it is appropriate to make comparisons with no differences between groups.

Please correct the data.

Page3, line123

In the results, only objective facts should be stated. Be concise and clear.

Page3, line123

Please describe the selection criteria for this study in the respondents. You need to prove that there is no selection bias.

Page3, line 132

Why did all three groups in this study have more female subjects than male subjects, even though it is said that there are more males in China? Is it suitable for investigating the health status of Chinese people mentioned in the objectives?

Page6, line189

I recommend that you include this in your results.

Page 7, line 202-207

I think it would be better to clearly state that Hubei residents are affected by strict lockdown policies and that Chinese people from overseas are affected by additional pressure from foreign societies on their health.

Page7, line 205

Please show me this report. Has such a thing been said before the spread of COVID-19?

Page7, line 222-227

It is unclear why the difference in response between governments leads to anxiety and what kind of mechanism it is. In order to clarify this explanation, it is necessary to unify information such as the area of residence and age of the overseas residents, and to specify the specific differences in government policy. Please correct it to a different content.

Page7 line 230-234

This discussion is too easy. There are many elderly people who exercise. As a social problem, the adverse effects of lack of exercise in the elderly are higher than those in the young.

Page7, line 240

Why did you design the questionnaire so that it would ask about age in these three age groups?

Page7, line249

I think it is important information that is polarized to a region. However, it was lacking in many ways to describe them. The concluding remarks are also not coherent.

Page8, line 251

As mentioned in the limitations, the present study does not reflect the Chinese people. Hence, I think this sentence is inappropriate.

Page8, line 262

It is difficult to propose these from this study. Do you have any suggestions for society based on the results of this research other than government policies? Please reconsider.

Page8, line282

References should be listed properly according to the form.

Page8, References

The number of authors listed differs from one reference to another, so it would be better to standardize the number of authors.

Table 2

What are the reasons why some of the numbers are not listed? Is it a problem with the number of subjects? Please explain.

Round 2

Reviewer 3 Report

The newly submitted version of the article is significantly revised compared to the previous (03-02-2021) version of the article. The authors took into account all three comments I made earlier and made appropriate corrections to the manuscript of the article. I also noticed that the authors took note of the comments made by other reviewers.

Reviewer 4 Report

Comment to the Author

Manuscript ID: ijerph-1117128

Title: Investigating the self-reported health status of Chinese populations at domestic and abroad during the COVID-19 pandemic

Page1, Line19

Did the pandemic affect only the Chinese? Has there been any comparison with other countries? This should be mentioned in the abstract. We can't understand the outline.

Page1, line19

Is this an appeal or proposal to the State? Is it really a task that can be accomplished?

Page1, line22

"You said "government in other countries", how are you going to achieve that? We can't see the future at all.

Page1, line24

We can't recognize any keywords from the title and abstract. We recommend that you consider this.

Page1, line28

Disclosure of cited sources is recommended.

Page1, line33

Is it still unresolved? Is this the current truth? Vaccines have been developed, but...

Page1, abstract line21-23

I think this sentence is inappropriate as a summary of the text. What does the government in other countries do?

Page2, line75

Isn't aboard misspelled as abroad?

Page2, line78-83

The two sentences appear to describe similar content. You should reconsider.

Page2, line98-99

This study did not compare the health care of Chinese people living in China and abroad, but rather “Hubei”, “outside Hubei”, and “oversea”. It would be better if it was stated correctly.

 Page4, line152

After all, I don't think this subgrouping is appropriate because it lacks good reason.

Page5, Table1

P value is listed, but I do not know which group there is a statistical difference between. The table should be redesigned to show the results of the Chi-tests.

Page5, Table 1

There were differences in the characteristics of the profiles in each of the three different regions. Did these differences in characteristics reflect the characteristics of the region? Or were there no differences in the characteristics of the three regions, and the differences just happened to appear in the target group?

Page6, Table3

Describe the results of the statistical analysis of the comparisons across the region described in line 156-158.

Page6, line212 and Page7, line228

Please change it to “odds ratio” instead of “odds”.

Page7, line238

 Overall, I feel that the effects of limitation is not taken into account.

Page8, line 268-273

Can "shortage of food supplies” and “lack of social interactions” cause PTSD? I think that "pass away of friends, relatives or someone close" may have a greater impact.

I think you are expressing your own opinion too much.

Page8, line287-294

It is difficult to understand what you want to convey from this sentence. Did this respondent live in the UK?

Page8, line295-303

I don't think this explanation is valid because telework tends to permeate the corona wreckage.

Page9, line314

Is it realistically possible?

Page9, line345
What was the rationale for determining that EQ-5D would be beneficial in this study?

Page9, line346-347

Does the view that there are regional differences in the results of self-reported health status due to the COVID-19 pandemic refer only to China? Are you suggesting that this may be the case in other countries as well?

Page9, line 356-357

I believe that this study investigated the variations of self-reported health status due to the COPID-19 pandemic and associated policies, and not whether the EQ-5D is useful for investigating health status due to the COPID-19 pandemic.

I do not think it should be written at the beginning of the Conclusion.

Introduction or Discussion

It was difficult to see what comparing the health care of Chinese people in three different regions using data from the early stages of infection would lead to. The situation is a little different now than in the early stages of infection. Is there anything that can be applied to the present?

Introduction or Discussion

COVID-19 is a global problem. Is there anything that can be adapted overseas as the Chinese data becomes clearer?

Abstract

Why don't the abstracts show specific results in numerical form?

Abstract line18-19

It is difficult to convey the content of this research with the current method of writing Abstract. Please compare the data and make it easy to understand.

Table

Basically, most of the tables are hard to recognize.

Author Response

Dear reviewer,

Thank you very much for taking the time and effort to review our manuscript and provide insightful guidance to improve it. The provided comments have been responded by us individually. We have addressed each concern and described the changes we made. The changes are marked in red in the revised manuscript.

Page1, Line19

Did the pandemic affect only the Chinese? Has there been any comparison with other countries? This should be mentioned in the abstract. We can't understand the outline.

Response: The pandemic is a global crisis that not only affects Chinese people but also influences people all around the world. However, our study focused on Chinese populations and we did not aim to make comparisons with other countries in this particular study. 

Page1, line19

Is this an appeal or proposal to the State? Is it really a task that can be accomplished?

Response: It is a policy implication for the government to considerr, based on the findings of our study. This particular study was not able to comment whether this task can be accomplished or not. 

Page1, line22

"You said "government in other countries", how are you going to achieve that? We can't see the future at all.

Response: Again, this is a policy suggestion based on the findings of our study. We cannot see the future but we can make suggestions aiming and hoping to make the future better.

Page1, line24

We can't recognize any keywords from the title and abstract. We recommend that you consider this.

Response: We may not agree with you in this regard. In both our title and abstract, we have the following keywords: “COVID-19” and “self-reported health status”. The study used EQ-5D to collect data on Chinese population health. EQ-5D was our main measurement and Chinese populations are our target populations, so we used “China”, “population health” and “EQ-5D” as our keywords too.

Page1, line28

Disclosure of cited sources is recommended.

Response: A citation has been added.

Page1, line33

Is it still unresolved? Is this the current truth? Vaccines have been developed, but...

Response: We did not state whether COVID-19 is unresolved or not. We only claimed that “COVID-19 pandemic is an unprecedented global crisis”.

Page1, abstract line21-23

I think this sentence is inappropriate as a summary of the text. What does the government in other countries do?

Response: In the last sentence of our abstract, we were making policy suggestions based on our findings. The summary of our study findings can be found in line 15-19.

Page2, line75

Isn't aboard misspelled as abroad?

Response: We have corrected this typo.

Page2, line78-83

The two sentences appear to describe similar content. You should reconsider.

Response: We have deleted the second sentence.

Page2, line98-99

This study did not compare the health care of Chinese people living in China and abroad, but rather “Hubei”, “outside Hubei”, and “oversea”. It would be better if it was stated correctly.

Response: Our study compared the health status of Chinese people in Hubei, outside Hubei and overseas. People in Hubei and outside Hubei were both living in China, so here we stated that this study was to “compare the health status of Chinese people both domestically and abroad”.

 Page4, line152

After all, I don't think this subgrouping is appropriate because it lacks good reason.

Response: The subgroup classification was determined by the design of our questionnaire. In our questionnaire, we classified age into 3 groups: under 40 years, 40-60 years, above 60 years. We considered this design as one of the limitations in our study and we stated that “at the time when we designed the questionnaire survey, we chose to collect age information by three age group (under 40 years, 41-60 years and above 60 years), without obtaining their exact ages, which restricted our statistical analysis”.

Page5, Table1

P value is listed, but I do not know which group there is a statistical difference between. The table should be redesigned to show the results of the Chi-tests.

Response: Here we performed Chi-square test for the three groups together to get a general idea about the differences between groups. Therefore, the p values are not for two groups but for three groups.

Page5, Table 1

There were differences in the characteristics of the profiles in each of the three different regions. Did these differences in characteristics reflect the characteristics of the region? Or were there no differences in the characteristics of the three regions, and the differences just happened to appear in the target group?

Response: We have responded this point in our last response to the reviewer. We collected our data using an online survey with a non-probability sampling strategy, and thus certain selection bias was inevitable. As a result, we obtained an unbalanced sample that cannot represent the general population. This was one of the limitations of our study. We controlled the demographic characteristics including gender, age, education, marital status, and chronic disease status in our regression models though.

Page6, Table3

Describe the results of the statistical analysis of the comparisons across the region described in line 156-158.

Response: We described that respondents from the three regions differed significantly in age, education, marital status, and chronic disease status in line 175-176. Regional differences in self-reported health status were presented in Table 2 and were described in line 189-198.

Page6, line212 and Page7, line228

Please change it to “odds ratio” instead of “odds”.

Response: They have been changed.

Page7, line238

Overall, I feel that the effects of limitation is not taken into account.

Response: The limitations of our study were discussed in line 311-335.

Page8, line 268-273

Can "shortage of food supplies” and “lack of social interactions” cause PTSD? I think that "pass away of friends, relatives or someone close" may have a greater impact.

I think you are expressing your own opinion too much.

Response: We have deleted “which may have made them develop post-traumatic stress disorder”.

Page8, line287-294

It is difficult to understand what you want to convey from this sentence. Did this respondent live in the UK?

Response: Here we only used UK as an example. We added in Results section that our overseas respondents were mainly from developed countries, while most of them were in Europe and North America.

Page8, line295-303

I don't think this explanation is valid because telework tends to permeate the corona wreckage.

Response: Here we only expressed our opinions to explain our findings. We felt that this explanation is not inappropriate.

Page9, line314

Is it realistically possible?

Response: This is a policy suggestion based on the findings of our study. We do not know if it is possible to achieve at this moment, but we can make suggestions aiming and hoping to make the future better.

Page9, line345
What was the rationale for determining that EQ-5D would be beneficial in this study?

Response: We have already deleted this sentence.

Page9, line346-347

Does the view that there are regional differences in the results of self-reported health status due to the COVID-19 pandemic refer only to China? Are you suggesting that this may be the case in other countries as well?

Response: We have added the evidence was about “Chinese people”.

Page9, line 356-357

I believe that this study investigated the variations of self-reported health status due to the COPID-19 pandemic and associated policies, and not whether the EQ-5D is useful for investigating health status due to the COPID-19 pandemic.

I do not think it should be written at the beginning of the Conclusion.

Response: We have already deleted this sentence about EQ-5D in Conclusion.

Introduction or Discussion

It was difficult to see what comparing the health care of Chinese people in three different regions using data from the early stages of infection would lead to. The situation is a little different now than in the early stages of infection. Is there anything that can be applied to the present?

Response: We made policy suggestions for the present and future based on the past experience.

Introduction or Discussion

COVID-19 is a global problem. Is there anything that can be adapted overseas as the Chinese data becomes clearer?

Response: We have added in the Conclusion that “The findings documented in this study are specific to China, but the implications may be considered in other countries.”

Abstract

Why don't the abstracts show specific results in numerical form?

Response: The main results of our study were that “People living in Hubei reported worse health status than those living outside Hubei but revealed better health status than overseas respondents”, which implies that “the pandemic as well as strict lockdown and outdoor restriction policies affected Chinese people’s health”. Differences in EQ-5D dimensions, VAS, utility scores and regression results were all numerical results but would be too much to be put in Abstract.

Abstract line18-19

It is difficult to convey the content of this research with the current method of writing Abstract. Please compare the data and make it easy to understand.

Response: In our abstract, we stated the background, the research aims, the methods, the respondents, the findings as well as the policy suggestions. We felt the abstract is a summary of our paper and we included the essential information about our study.

Table

Basically, most of the tables are hard to recognize.

Response: We have made our efforts to present our results as clear as possible in the Tables. We have described the results in the tables in the main text as well.